# The Effect of Glucose on the Interaction of Bisphenol A and Bovine Hemoglobin Characterized by Spectroscopic and Molecular Docking Techniques

**DOI:** 10.3390/ijms241914708

**Published:** 2023-09-28

**Authors:** Xianheng Li, Huan Li, Keqiang Lai, Junjian Miao

**Affiliations:** 1College of Food Science and Technology, Shanghai Ocean University, No. 999 Hucheng Huan Road, Lingang New City, Shanghai 201306, China; d210300073@st.shou.edu.cn (X.L.); 13635788154@163.com (H.L.); 2Engineering Research Center of Food Thermal-Processing Technology, Shanghai Ocean University, Shanghai 201306, China

**Keywords:** glucose, bisphenol A, bovine hemoglobin, spectroscopic techniques, molecular docking

## Abstract

The interaction mechanism of hemoglobin (Hb) with bisphenol A (BPA) in diabetic patients and the difference with healthy people have been studied using spectroscopic and molecular docking techniques at several glucose (Glc) concentration, with bovine hemoglobin (BHb) instead of Hb. It is found that Glc can interact with BHb–BPA and affect its molecular structure, resulting in an altered microenvironment for tyrosine (Tyr) and tryptophan (Trp) in BHb–BPA. It is also found that Glc can bind to BHb alone, and its effect on the molecular structure of BHb is weaker than that on the structure of BHb in BHb–BPA complex. The results of circular dichroism (CD) and Fourier transform infrared spectroscopy (FTIR) indicate that Glc causes an increase in the content of the α-helix and a decrease in that of the β-sheet of BHb–BPA by 1.5–1.9% and 3.1%, respectively. The results of molecular docking show that Glc binds to BHb–BPA through hydrogen and hydrophobic bonds, and the position of binding differs from that of Glc binding to BHb alone, which may be attributed to the fact that BPA affects the protein molecular structure of BHb and has an effect on the binding of BHb to Glc. This study provides some theoretical basis for the mechanism of BPA toxicity in vivo for people with different blood glucose levels.

## 1. Introduction

As one of the most produced chemicals in the world [1], BPA is polymerized to produce a very hard plastic called polycarbonate. Such plastics are used in the inner walls of food cans, water pipes, baby bottles, industrial facilities, medical devices, etc. [2,3]. Due to the extensive use of plastics in people’s lives, BPA can enter human body through different routes such as the digestive system, respiratory system and skin [4,5]. Numerous studies have shown that BPA is an endocrine analog with multi-organ toxicity that has serious effects on reproduction and the nervous system, and it can affect the development of offspring, as well as cause health hazards such as metabolic disorders, cancer and cardiovascular disease [6,7,8]. Researchers have detected BPA residues in body fluids, blood and tissues at different concentrations [5]. These BPA residues may interact with some proteins in the blood undermining the function of the human body [9].

Hb is the major functional protein in erythrocytes in human blood and has been associated with many diseases such as leukaemia, anemia and heart disease [10]; it plays an important physiological role in oxygen transport, carbon dioxide transport and blood pH regulation [11,12]. BHb has been used as a substitute for Hb in scientific research due to its high homology [13]. Several studies have found that BPA can bind to BHb and obtain binding constants and binding sites for both [14]. Glc is a monosaccharide known as “blood sugar”. It has been reported that Glc is able to bind to Hb as a ligand [15]. Mutational effects of molecules indicate that when more than one ligand binds to a protein, the binding of the first ligand may result in a change in the structure of the protein, which in turn affects the binding of the other ligands [16]. Therefore, it can be hypothesized that the binding of Glc to BHb may affect the interaction of BPA with BHb. The aim of this work is twofold: (1) to investigate the effect of Glc on the interaction of BHb and BPA; and (2) to further examine the working mechanism of the effect. The results of the study will contribute to a better understanding of the effect of Glc on BHb–BPA in the bodies of healthy and diabetic individuals.

## 2. Results and Discussion

### 2.1. Absorption Spectroscopy

UV-Vis absorption spectroscopy techniques are commonly used to study structural changes in proteins and the formation of protein–ligand complexes. Appendix A and Figure 1A shows the results of the interaction between BHb and BPA. The data in the figure show two absorption peaks at 280 nm and 405 nm for BHb. The absorption peak of aromatic amino acids (Trp, Tyr and Phe) is reached at 280 nm, with that of the porphyrin-Soret band of BHb at 405 nm [17,18]. With the increase in BPA concentration, the intensity of the absorption peaks of the BHb–BPA complex samples near 280 nm increased and had a weak redshift (Figure 1A). The reason for this phenomenon may come from two aspects: on the one hand, the increase in the absorption strength mainly originates from the increase in the concentration of BPA, which can be observed by comparing Appendix A and Figure 1A. On the other hand, the interaction between BPA and BHb leads to a change in the microenvironment of aromatic amino acids in the protein, which slightly enhances UV absorption and causes a weak redshift [13,14]. However, the peak at 405 nm did not change significantly, showing that there is no direct interaction of BPA with the porphyrin ring of BHb. However, the peak at 405 nm did not change significantly, showing that there is no direct interaction of BPA with the porphyrin ring of BHb.

To investigate the effect of Glc on the BPA–BHb interaction, the experiments were repeated in phosphate buffers containing Glc at concentrations of 4 mM, 7 mM and 11 mM. According to the Figure 1B results, it can be seen that the absorbance of the complexes further increased when BHb–BPA was treated with higher concentrations of Glc. This phenomenon can be attributed to two factors. First, the binding of Glc molecules to BPA alters the structure of BHb; this hypothesis is based on the phenomenon of allosteric effect, where the binding of the first ligand may cause structural changes in the protein, leading to a change in the binding site when the second ligand binds to the protein [16]. The interaction between BHb and Glc was investigated under similar experimental conditions in the absence of BPA to examine the effect of Glc molecules on the structure of BHb. Figure 1C results showed that Glc increased the absorbance of BHb at 280 nm, which could be a result of a change in the microenvironment of Trp residues [19]. These residues are usually located in the internal region of BHb, and the change in the microenvironment of Trp residues may be due to structural changes in the protein caused by the interaction of the protein with Glc. Another possibility is that Glc molecules bind to BPA molecules and enhance their binding to BHb. To further test this hypothesis, the interaction between BPA and Glc was investigated in the absence of BHb. According to Figure 1D results, it was found that Glc increased the absorbance of BPA and this effect was stronger when BPA was treated with higher concentration of Glc. This suggests that BPA and Glc molecules may interact with each other and both explanations seem possible. In order to better reveal the cause of Glc on BHb–BPA, we conducted fluorescence experiments. In conclusion, Glc causes an increase in the absorption of BPA–BHb at 280 nm and it affects the structure of the BPA–BHb complex.

### 2.2. Steady-State Fluorescence

According to a related study, BHb can emit endogenous fluorescence, and it is the tyrosine and tryptophan residues that play a major role [20]. In this study, the fluorescence spectra of different concentrations of BPA interacting with BHb were investigated. As shown in Figure 2A, a fluorescence emission peak was recorded near 330 nm after excitation at 280 nm. From the results in Figure 2A, it can be seen that BPA and BHb bonded to each other. The molecular structure of BHb was altered, and the peak fluorescence of BHb increased by 2% with increasing BPA concentration, and a weak blue shift occurred, which was produced by the binding of BPA to BHb, resulting in the structural alteration of BHb and the enhancement of the hydrophobicity to the Tyr and Trp microenvironments inside BHb [21,22]. The results from Figure 2B showed that as the concentration of Glc increased from 4 mM to 7 mM to 11 mM, the increase in the peak fluorescence intensity of BHb–BPA increased by 10.9%, 16.8% and 28.7%, respectively, and a weak blue shift occurred. This may be due to the interaction of Glc with tryptophan and tyrosine in BHb or with BPA, which affects the fluorescence emission signal of BHb–BPA [23]. To investigate the changes in the interaction between Glc and BHb, the peak BHb fluorescence intensity increased with increasing Glc concentration from 4 mM to 7 mM to 11 mM. The peak BHb–BPA fluorescence intensity increased by 2.0%, 5.9% and 12.7%, and the comparison of the magnitude of the two changes revealed that the effect of Glc on the structure of BHb was weaker than that of Glc on the structure of BHb–BPA, which could be attributed to the interaction between Glc and BPA affecting the change in the fluorescence intensity of BHb–BPA. This result is consistent with the UV absorption spectroscopy result that Glc interacts with both BHb and BPA and thus affects BHb–BPA. In general, the steady-state fluorescence spectra reflect the changes in the overall fluorescence intensity of the molecule. In order to analyze the interactions between Glc and the amino acids in the BHb–BPA complex in more depth. This study investigated the effects of different concentrations of Glc on the structure of BHb–BPA by means of simultaneous fluorescence spectroscopy.

### 2.3. Synchronized Fluorescence Spectroscopy

In synchronous fluorescence spectroscopy, the fluorescence spectrophotometer scans the excitation and emission monochromator with a constant wavelength difference. Due to these properties, the method is more sensitive and accurate than steady-state fluorescence. The structural information of tyrosine and tryptophan was obtained when the difference between excitation and emission light intensities was 15 and 60 nm, respectively [24,25]. Figure 3A shows the changes in fluorescence values of tyrosine emission peaks caused by incubating BHb with different concentrations of BPA. It can be seen that as the concentration of BPA increased, the emission spectra of BHb changed markedly and the fluorescence peaks produced a weak blue shift, indicating that BPA had an effect on Tyr in BHb. When the difference between the excitation and emission wavelengths is 60 nm, the resulting emission spectra would be associated with Trp residues. Figure 3B shows the BHb was incubated with different concentrations of BPA produced a blue shift in the Trp fluorescence peak, but the change was not significant, probably because the Trp microenvironment in BHb was not significantly affected by BPA. In order to investigate the mechanism of the effect of different Glc on BHb–BPA, the synchronized fluorescence spectra of BHb–BPA complexes after incubating with different concentrations of Glc were detected. Figure 3C,D are the synchronized fluorescence spectra representing Tyr and Trp, respectively, which showed that the fluorescence peaks of both Tyr and Trp were elevated with the increase in the concentration of Glc, suggesting that the interaction between Glc and BHb interacted and affected the distribution of Tyr and Trp in BHb. It has been reported that the fluorescence changes of Hb are very closely related to Trp37 [26]. From the results of molecular docking results for Glc and BHb, it can be seen that Trp37 in BHb interacts hydrophobically with Glc, so it may be due to the binding of Glc to BHb affecting the microenvironments of Trp37 in BHb. The results by UV spectroscopy and steady-state fluorescence spectroscopy suggest that Glc may affect the structure of the complex of BHb–BPA. In order to further investigate the mechanism of Glc on BHb–BPA, the effects of different concentrations of Glc on BHb–BPA were investigated by simultaneous fluorescence spectroscopy, and Figure 3E,F corresponded to Tyr and Trp, respectively. The fluorescence peaks of both Tyr and Trp were gradually increased with the increase in Glc concentration. It is noteworthy that the increase in the peak fluorescence of Tyr with increasing Glc concentration in BHb–BPA in Figure 3C is similar to that of Tyr with increasing Glc concentration in BHb in Figure 3E, whereas the peak fluorescence of Trp with increasing Glc concentration is 23.5% in BHb–BPA in Figure 3D, and 13.8% in BHb in Figure 3F. This may also be the reason why the effect of Glc on the fluorescence peak of BHb in the steady-state fluorescence spectra was weaker than that of Glc on the fluorescence peak of BHb–BPA.

### 2.4. Circular Dichroism

CD spectroscopy can be used to study the secondary structure of proteins, and it allows for sufficient data to estimate the proportion of the protein structure that involves α-helices, β-folds or regular coils [27,28]. To further investigate the effect of Glc on the conformation of the BHb–BPA complex, a CD spectral analysis was conducted. The spectrum of pristine BHb exhibits a typical profile with two values at 208 and 222 nm that are characteristic bands of the α-helix [29], the 208 nm band points to the π–π* transition of the α-helix, and the 222 nm band corresponds to the α-helix and the n–π* transition of the random structure [17]. With the addition of BPA, we found a decrease in the CD values in the CD spectra of BHb. The addition of Glc had little effect on the CD value of BHb. When Glc was added to the BHb–BPA mixture, there was a modest decrease in the CD value of BHb–BPA, but no significant shift was noted in the peak position of the CD spectrum of BHb after the addition of BPA and Glc, which suggests that BPA and Glc induced a conformational change in the native protein, which resulted in a change in the α-helix of BHb. Figure 4 shows the CD spectra of BHb. The content of the α-helix in BHb was calculated from the CD spectra via Equations (1) and (2) [30], with Equation (1) as:(1)$${MRE}_{208}=\frac{ObservedCD\left(mdeg\right)}{C_{p}nl}\times 10$$

In Equation (1), MRE_208_ is the mean residue ellipticity at 208 nm (deg cm^2^ dmol^−1^), C_P_ is the molar concentration of the protein, n is the number of amino acid residues (574), and l (cm) is the path length of the unit. The α-helix content was calculated from the MRE value at 208 nm [24] and Equation (2) is:(2)$$\alpha –Helix\left(\%\right)=\frac{-{MRE}_{208}-4000}{33,000-4000}\times 100$$

In Equation (2), 4000 is the MRE value at 208 nm for the β-type and irregularly coiled conformations. 33,000 is the MRE value at 208 nm for the α-helix.

**Figure 4 ijms-24-14708-f004:**
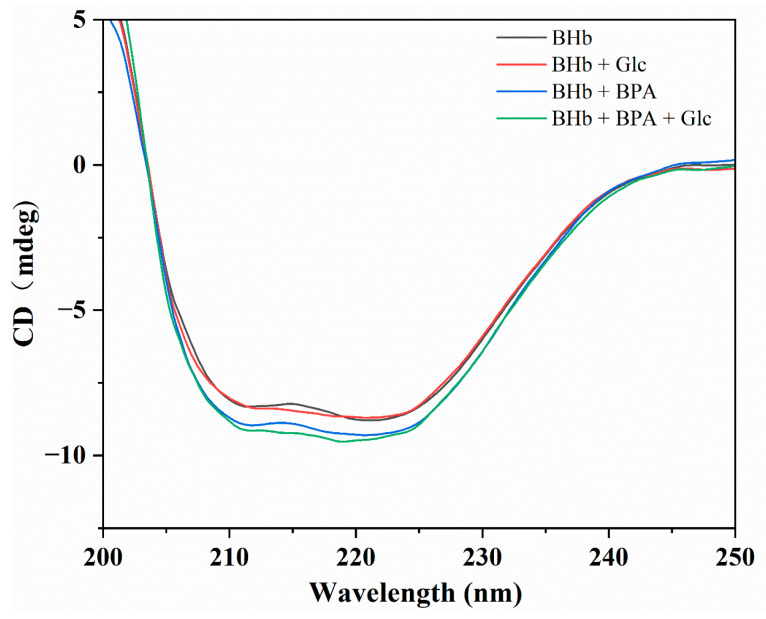
The CD spectra of BHb, BHb + Glc, BHb + BPA and BHb + BPA + Glc native. The concentration of BHb is 1 μM and the BPA concentration is 10 μM. The concentration of Glc is 11 mM.

From Table 1 it can be seen that the content of α-helix in BHb is 28.7% and after incubation with Glc, the content of α-helix in BHb increased to 29.7%, which is almost unchanged compared to BHb. In contrast, after incubation with BPA, the α-helix content of BHb increased to 32.4%, and after incubation of the BHb–BPA complex with Glc, the α-helix content of the protein increased slightly to 33.9%. The above analysis shows that BPA affected the secondary structure of BHb, which increased the content of the α-helix of BHb by 3.7%, and the content of the α-helix of BHb–BPA was increased by 1.5% after Glc incubated with BHb–BPA. It is clear that the effect of Glc on the secondary structure of BHb is small, and at the same time, Glc also affects the secondary structure of the BHb–BPA complex.

### 2.5. Fourier Infrared Spectroscopy Studies

FTIR is a well-established technique that has been used by researchers in many studies to analyze changes in the secondary structure of proteins [31,32]. Figure 5 and Figure 6 and Table 2 show the changes in the secondary structure of BHb after its interaction with Glc and BPA, which can help analyze the effect of Glc and BPA on the structure of BHb and to validate the results of circular dichroism. In order to investigate the effect of BPA binding to BHb on the secondary structure of BHb, the changes in the secondary structure of BHb were detected by FTIR after incubation of a final concentration of 1 μM of BHb and 10 μM of BPA for 30 min, and compared with the same concentration of BHb with no BPA added. The experimental results show that after the addition of BPA, the α-helix of BHb increased by 3.0%, the content of β-sheet decreased by 3.6%, and the content of β-turn, antiparallel β-sheet and random coil changed weakly, which may be due to the hydrogen bonding and hydrophobic interactions between BPA and BHb molecules that affected the secondary structure of BHb. In order to investigate the effect of Glc on the secondary structure of BHb in BHb–BPA, the changes in the secondary structure of BHb were detected after the addition of Glc incubation in BHb–BPA solution for 30 min, and it was found that the content of α-helix in BHb–BPA was increased by 1.9% and the content of β-sheet was decreased by 3.1% by Glc, which indicates that the effect of Glc on the secondary structure of BHb–BPA is not significant. This indicates that Glc has a definite effect on the secondary structure of BHb–BPA. In order to investigate whether the reason for the effect of Glc on the structure of BHb–BPA is related to the interaction between Glc and BHb, the secondary structure of BHb was examined after adding Glc incubation for 30 min to BHb solution without BPA, and it was found that Glc increased the content of α-helix and decreased the content of β-sheet by 1.4% and 2.0%, which is a change of 1.9% and 3.1%, respectively. The results showed that Glc increased the α-helix content of BHb by 1.4% and decreased the β-sheet content of BHb by 2.0%, which was similar to the trend of the effect of Glc on the secondary structure of BHb–BPA, and the results of FTIR and circular dichroism on the α-helix content of BHb were different but has the same trend, which may be due to the fact that circular dichroism is highly sensitive to the changes in the secondary structure of proteins, especially when a bright synchrotron radiation (SR) source is used and some chemicals that may introduce errors were generated [33]. The results obtained by other researchers also found that the α-helix content obtained by circular dichroism detection of the secondary structure of proteins was higher than the α-helix content obtained by FTIR detection, which is consistent with the results of this study [34]. Taken together, BPA and Glc affect the secondary structure of BHb, resulting in an increase in the α-helix content and a decrease in the β-sheet content of BHb, and Glc also affects the changes in the secondary structure of BHb–BPA. These effects may be due to the binding of Glc to BHb.

### 2.6. Molecular Modeling Study

Computer simulation studies can contribute to a deeper understanding of the interactions of BPA and Glc with BHb at the molecular level and to clearly characterize the effects of both on the secondary structure of BHb. In this study, molecular docking was employed to calculate the binding energy of protein–ligand complexes with the help of Autodock. Overall, the higher the binding energy, the more suitable the complex formed. According to the theoretical calculations, the best results of BPA and Glc binding to BHb are shown in Figure 7 and Figure 8. In order to investigate the mechanism of action of Glc binding to BHb–BPA with each other, molecular docking results of Glc binding to BHb–BPA were carried out with results shown in Figure 9. The binding energies of BPA and Glc binding to BHb were −5.86 kcal and −4.11 kcal, respectively. For analysis of the molecular docking results of Glc with BHb–BPA, the top 10 docking results with the lowest binding energies were obtained based on binding energy ranking (Appendix A), and then the docking result with the lowest binding energy of −4.1 kcal was selected. This shows that BPA is more likely to bind to BHb than Glc, and has a greater effect on the structure of BHb, which is consistent with the results obtained from spectroscopic experiments.

Figure 7 gives the results of molecular docking of BHb with BPA. Figure 7A shows that BPA can be bound to the inner molecule of BHb. The amino acids located 4.00 Å from BPA within BHb are also reflected in Figure 7A, and these amino acids reflect the profile of BPA binding to the central hydrophobic cavity of BHb. It is noteworthy that the results of Figure 7B obtained from the Ligplot program indicate that these residues (Ser35A, Phe36A, Pro37A, Ala135B, Tyr145D) contribute to the formation of the hydrophobic cavity through hydrophobic interaction. Most importantly, Thr38A, His146D and Asn139B were in a suitable position to form intermolecular hydrogen bonding interactions with BPA. The results of fluorescence spectroscopy showed that the binding of BPA to BHb caused the fluorescence intensity of BHb to increase, which on the one hand could be attributed to the formation of hydrogen bonding between Tyr145D and BPA in BHb, and on the other hand, based on the results of FTIR and CD spectroscopy results, BPA causes changes in the secondary structure of BHb, affecting the distribution of amino acids and, thus, the fluorescence intensity of BHb.

In order to investigate the interaction between Glc and BHb, the molecular docking technique was applied in this study to predict the interaction mechanism of the two, and the results are shown in Figure 8. It can be seen from Figure 8A that Glc can bind to BHb, Figure 8A shows the amino acids within the BHb at a distance of 4.00 Å from Glc, and most of these amino acids interact with Glc, essentially reflecting the profile of Glc’s hydrophobic cavity in BHb. In order to better examine the interaction mechanism between the two, the Ligplot program was launched to plot the two-dimensional results of the interaction between Glc and BHb, and the results are shown in Figure 8B. The results showed that Glc was able to produce hydrophobic interactions with some amino acid residues (Ser138C, Trp37B, Pro95C, Ala130A) and form hydrogen bonds as well with some amino acid residues (Thr134C, Thr137C, Tyr140C, Lys127A, Arg141C). It has been suggested that Trp37 is associated with the fluorescence of BHb, and the fluorescence of Glc with BHb may be related to the hydrophobic interaction of Glc with Trp37B and the hydrogen bonding of Glc with Tyr140C.

In order to study the effect of Glc on BHb–BPA, BHb was molecularly docked with BPA, and then Glc was molecularly docked with BHb–BPA molecular complex, and the final results obtained are shown in Figure 9. It can be seen that Glc can bind with BHb–BPA, but the binding regions of Glc and BHb–BPA do not intersect with each other. And as for the details of the mutual binding of Glc and BHb–BPA, Figure 9A shows the amino acid molecules within BHb–BPA that are at a distance of 4.00 Å from Glc. The results of molecular docking are sometimes often easier to understand with 2D images compared to 3D images. A 2D image of the interaction of Glc with BHb–BPA was plotted by Ligplot program in Figure 9B, in which it can be seen that some amino acid residues (Leu48D, Gly46D, Glu43D, Arg92A, Gln39D) form hydrogen bonding interactions with Glc, while some amino acid residues (Asp47D, Ser49D, Arg40D) form a hydrophobic structure with Glc. It is noteworthy that Glc is not surrounded by Tyr and Trp, while in fluorescence spectra it was found that Glc affects the fluorescence properties of BHb–BPA, and the reason for this result may come from two aspects: first, some of the BHb that is not bound to BPA binds to Glc, and the fluorescence spectra and molecular docking results of Figure 8 show that Glc interacts with Trp37B and hydrophobic interactions with Tyr140C, which led to changes in the fluorescence of BHb; second, according to the results of FTIR and CD spectroscopy Glc causes changes in the secondary structure of BHb–BPA, so it is possible that the binding of Glc to BHb–BPA causes changes in the amino acid microenvironment of Tyr and Trp due to changes in the secondary structure of BHb–BPA, resulting in changes in the fluorescence value of BHb–BPA.

## 3. Materials and Methods

### 3.1. Materials

BPA (≥99%) and BHb (≥99%) are both purchased from Shanghai Aladdin Reagent Co. BHb was dissolved in pH 7.4 buffer to form a 1 mol L^−1^ solution, which was then stored at 4 ^◦^C and diluted as needed. BPA (10 mol L^−1^) stock solution was prepared by dissolving BPA in 10 mL of methanol. The pH was controlled using 0.1 M phosphate buffer (a mixture of NaCl, KCl, KH_2_PO_4_ and Na_2_HPO_4_). The KH_2_PO_4_ and Na_2_HPO_4_ were of analytically pure grade and were purchased from Sinopharm Co., Ltd. (Shanghai, China).

### 3.2. Spectroscopic Studies

#### 3.2.1. Ultraviolet (UV) Spectroscopy

UV spectrophotometry is often used to detect changes in the structure of proteins and compounds [27,35]. In this experiment, the absorption spectra of different concentrations of BPA interacting with BHb were determined in the range of 200–600 nm. The molar ratios of BHb to BPA were 1:0, 1:5, 1:10, 1:15 and 1:20. The concentration of BHb in the assayed solution was 1 μM. In order to investigate the effect of Glc on BHb–BPA, the concentration of BHb was 1 μM. To study the effect of Glc on the BHb–BPA complex, different concentrations of Glc were added to the 1:10 molar ratio of BHb to BPA solution, mixed thoroughly and incubated at room temperature for 30 min, in which the final Glc concentrations in the solutions to be tested were 0, 4, 7 and 11 mM, respectively. The effect of Glc on the BHb–BPA complex was investigated by detecting the change in absorption peaks of the benzene ring at 278 nm and that on protein structure by recording changes in the absorption peak of the porphyrin ring at 410 nm [36].

#### 3.2.2. Steady-State Fluorescence Spectra

BHb consists of two α-chains and two β-chains, with each α-chain containing three Tyr residues and one Trp residue, and each β-chain containing two Tyr residues and two Trp residues [37]. All of these residues can absorb or emit fluorescence, so their interactions with other molecules can be analyzed by detecting changes in the fluorescence of BHb [38]. The BHb to BPA ratio, Glc concentration, and incubation time were the same as in the UV test experiments. All experiments were performed at room temperature. The fluorescence spectrometer model was Shimadzu RF-5301PC. The width of the slit was 3 nm for Ex and 5 nm for Em. The excitation wavelength was chosen to be 280 nm. The emission wavelengths were recorded between 300 and 500 nm, and all the data displayed are the average of three repetitions of the measurements.

#### 3.2.3. Synchronized Fluorescence Spectroscopy

Synchronized fluorescence spectra of BPA with BHb and Glc at 298 K were investigated for different Glc concentrations, with slit widths of 3 nm for Ex and 5 nm for Em. The wavelength intervals (Δλ) between the excitation wavelength (λex) and the emission wavelength (λem) were set at Δλ = 15 nm and Δλ = 60 nm, respectively (Δλ = λem − λex). The excitation wavelengths at Δλ = 15 nm and Δλ = 60 nm are 235 nm and 220 nm, respectively. The detection ranges were 250–400 nm and 280–400 nm, respectively, and the detection substances corresponding to the two wavelength intervals were Tyr and Trp, respectively [39].

#### 3.2.4. Circular Dichroism Spectroscopy

CD spectral measurements were performed on a JASCO-J-1500 spectropolarimeter (Tokyo, Japan). The scan rate was 100 nm·min^−1^ and the band width was selected to be 1 nm. CD measurements of the samples were carried out in the range of 190–250 nm, and three scans were averaged for each CD spectrum. CD samples were prepared at fixed concentrations of BHb (1 μM) with BPA (10 μM) and Glc (11 mM).

#### 3.2.5. FTIR Spectra

Fourier transform infrared spectroscopy is an effective method used to characterize changes in protein secondary structure. Among all the amide bands of BHb, the amide I band (1700–1600 cm^−1^) is associated with C=O stretching and the amide II band (1600–1500 cm^−1^) with C-N stretching and N-H bending. Many researchers have focused mainly on the amide I band because it is more sensitive to secondary structure changes in proteins than the amide II band [31,40]. In order to investigate the effect of Glc on the secondary structure of BHb–BPA, we obtained FTIR spectra by analyzing BHb and BHb interactions with BPA and Glc using an FTIR spectrometer (Thermo Fisher Scientific, Waltham, MA, USA) equipped with an ATR accessory. The spectral region analyzed was 4000–400 cm^−1^, the resolution was set at 4 cm^−1^ and the scan period was at 64. The above spectral region was deconvoluted by second-order derivatives in the spectral region of 1700–1600 cm^−1^ and adjusted to correspond to the α-helix (1660–1649 cm^−1^), β-folding (1638–1610 cm^−1^), β-turn (1680–1660 cm^−1^), β-antiparallel (1692–1680 cm^−1^) and irregular curl (1648–1638 cm^−1^) peaks, and finally, the areas of all component bands assigned to a given conformation were summed and divided by the total area [24]. Samples were prepared with fixed concentrations of BHb (1 μM) with BPA (10 μM) and Glc (11 mM).

### 3.3. Study of Molecular Docking Techniques

In order to recognize the effect of Glc on the action of BPA and BHb at the molecular level, the sites of interaction of Glc and BPA with BHb and Glc with BHb–BPA are predicted in this study through molecular docking. The structure of BHb was obtained from the RCSB Protein Data Bank (PDB ID: 1G09) [20]; other systems in the crystal were removed before docking and essential hydrogen atoms were added with the help of AutoDock tool. The 3D structures of the ligands BPA and Glc were obtained in PubChem. Charges were added to the ligand molecules, rotatable bonds were identified and nonpolar hydrogen atoms were assigned. Boxes of size 126 × 126 × 126 Å with a grid spacing of 0.558 Å were created using the AutoGrid program. Default values given by the program were used for other parameters. Docking calculations were performed using the Lamarckian genetic algorithm (LGA). The number of runs was set to 250,000. Three-dimensional maps of protein and ligand molecular interactions were viewed using PYMOL [41]. The molecular docking software used was AutoDock 4.2 [42].

## 4. Conclusions

In this study, the effect of glucose on the interaction of bisphenol A and bovine hemoglobin has been characterized by spectroscopic and molecular docking techniques. The results show that Glc can interact with BHb and also have intermolecular interactions with BPA, and its effect on BHb–BPA may be due to the combined effect of the allosteric effect of BHb induced by BPA and the intermolecular interactions between Glc and BPA. The results of fluorescence spectroscopy revealed that the peak fluorescence intensity of BHb–BPA gradually increased from 10.9% to 28.7% as the concentration of Glc increased from 4 to 11 mM, respectively. After incubating Glc alone with BHb, the peak fluorescence intensity of BHb increased from 2.0% to 12.7% with increasing Glc concentration. This variation was weaker than the effect of Glc on the fluorescence change in BHb–BPA. It shows that BPA affects the structure of BHb, which leads to more obvious changes in the molecular structure of the complex after the binding of Glc and BHb–BPA. The peak fluorescence of Trp in BHb–BPA increased from 23.5% with the increase in Glc concentration, while the peak fluorescence of Trp in BHb increased from 13.8%. Incubation of Glc with BHb–BPA induced changes in its secondary structure, resulting in a 1.5–1.9% increase in the content of the α-helix and a 3.1% decrease in the content of the β-sheet of BHb–BPA. The molecular docking results show that Glc and BPA can alter the protein molecular structure of BHb through hydrogen bonding and hydrophobic interaction. Hydrogen and hydrophobic bonds likewise play a major role in the binding between Glc and BHb–BPA complexes. This study provides some theoretical basis for the mechanism of BPA toxicity in vivo for people with different blood glucose levels.

## Figures and Tables

**Figure 1 ijms-24-14708-f001:**
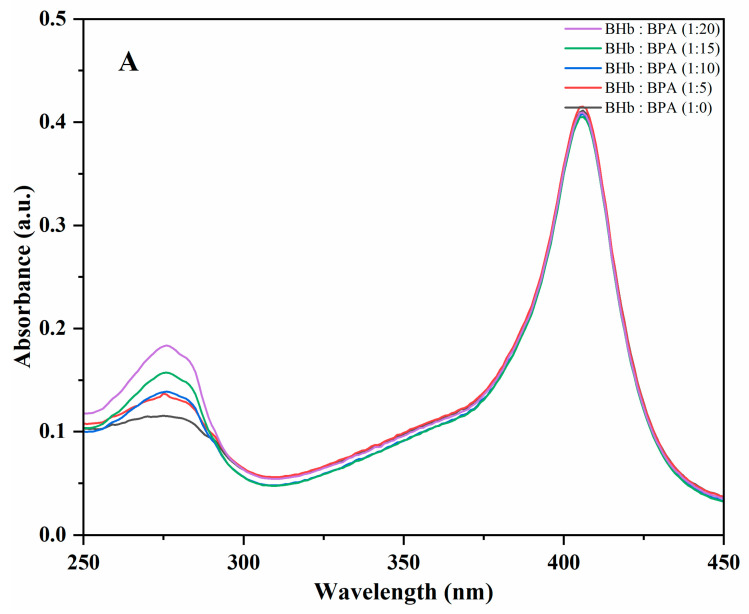
(**A**) UV spectra of BHb treated with different concentrations of BPA. The BHb was 1 μM and the concentration ratios of BHb:BPA were 1:0, 1:5, 1:10, 1:15, 1:20. (**B**) UV spectra of BHb and BPA in glucose solutions at different concentrations. concentration of BHb is 1 μM and BPA concentration is 10 μM. (**C**) UV spectra of BHb in glucose solutions at different concentrations; concentration of BHb is 1 μM. (**D**) UV spectra of BPA in glucose solutions at different concentrations; concentration of BPA is 10 μM.

**Figure 2 ijms-24-14708-f002:**
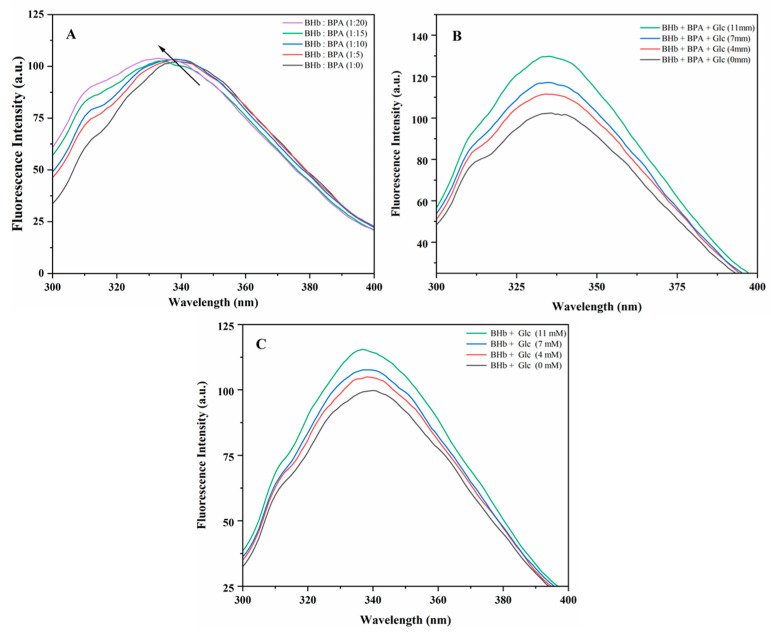
(**A**) Fluorescence spectroscopy of BHb treated with different concentrations of BPA. The BHb was 1 μM and the concentration ratios of BHb: BPA were 1:0, 1:5, 1:10, 1:15, 1:20. (**B**) Fluorescence spectroscopy of BHb and BPA in glucose solutions at different concentrations. The concentration of BHb is 1 μM and the BPA concentration is 10 μM. (**C**) Fluorescence spectroscopy of BHb in glucose solutions at different concentrations. The concentration of BHb is 1 μM.

**Figure 3 ijms-24-14708-f003:**
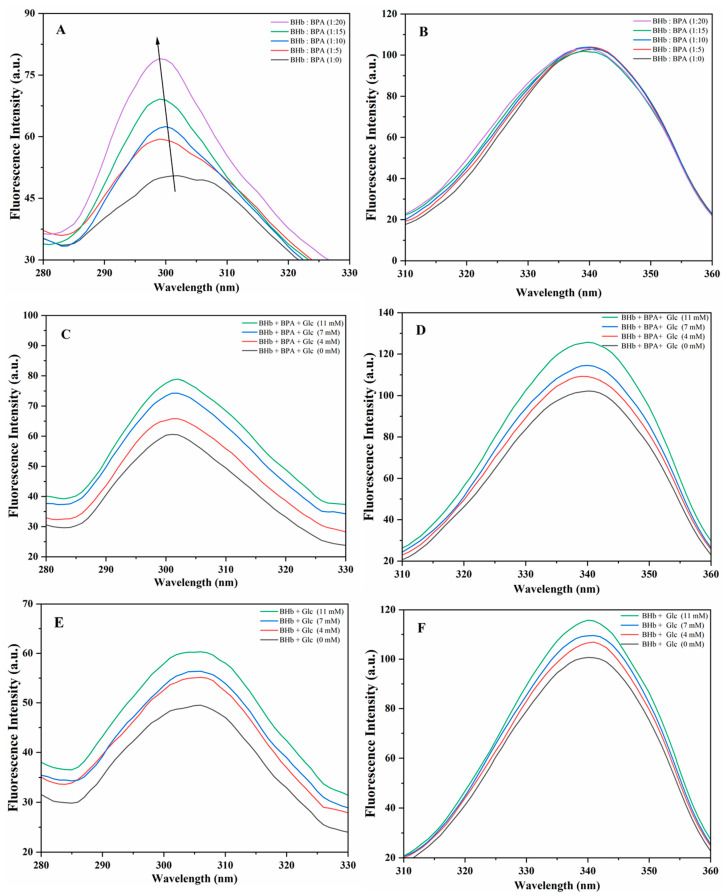
Synchronous fluorescence spectra of the interaction of BHb with BPA and Glc. (**A**) The Tyr residues of BHb (1 μM) in the presence of different concentrations of BPA (Δλ = 15 nm). (**B**) The Trp residues of BHb (1 μM) in the presence of different concentrations of BPA. (Δλ = 60 nm). (**C**) The Tyr residues of BHb (1 μM) after incubation with BPA (10 μM) in the presence of different concentrations of Glc (Δλ = 15 nm). (**D**) The Trp residues of BHb (1 μM) were incubated with BPA (10 μM) in the presence of different concentrations of Glc (Δλ = 60 nm). (**E**) The Tyr residues of BHb (1 μM) incubated with different concentrations of Glc (Δλ = 15 nm). (**F**) The Trp residues of BHb (1 μM) were incubated with different concentrations of Glc (Δλ = 60 nm).

**Figure 5 ijms-24-14708-f005:**
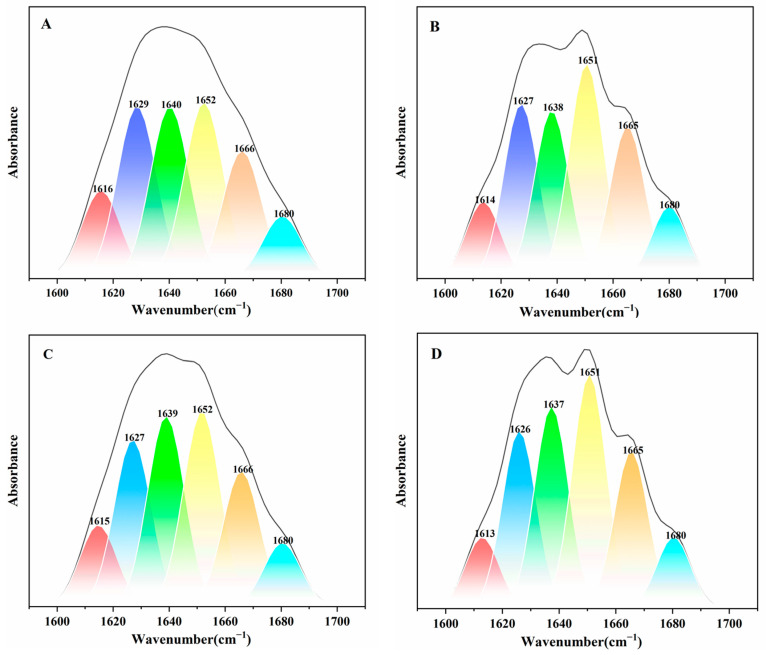
The curve fitting amide I band by second derivative resolution of BHb (**A**), BHb–BPA (**B**), BHb–Glc (**C**), BHb–BPA–Glc (**D**) system. The concentration of BHb, BPA and Glc are 1 μM, 10 μM and 11mM, respectively.

**Figure 6 ijms-24-14708-f006:**
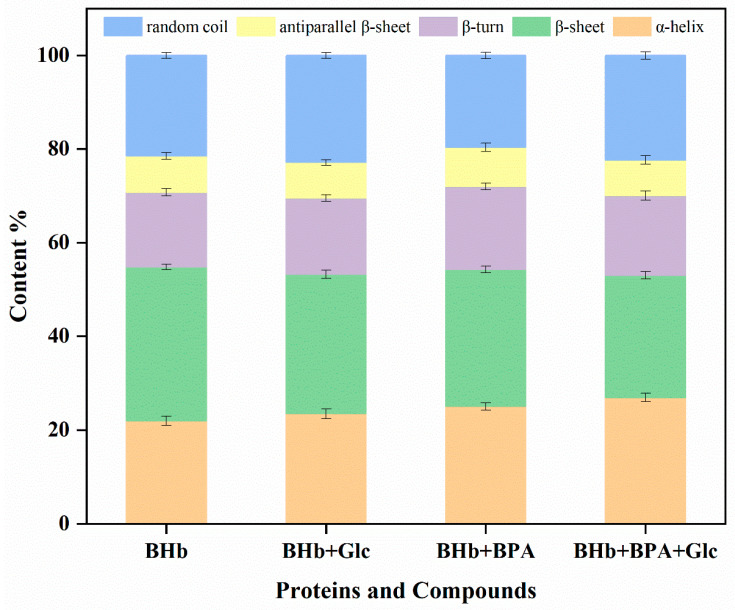
Content (%) of secondary structure of BHb treated with different concentrations of BPA and Glc.

**Figure 7 ijms-24-14708-f007:**
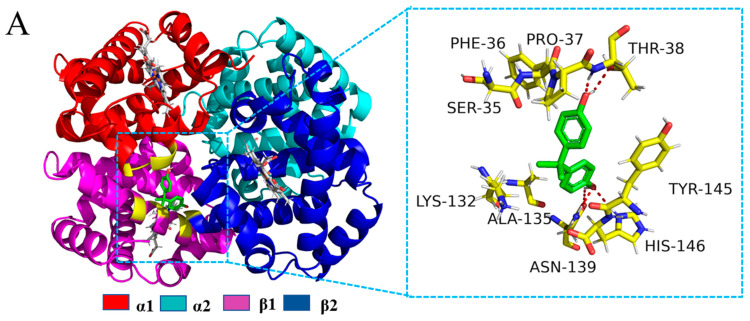
(**A**) Cartoon of the combination of BHb and BPA. And the amino acid residues of BHb surrounding BPA within 4.00 Å (red dashed lines represent hydrogen bonds). (**B**) The 2D molecular interaction plot for BPA surrounded by contacting receptor residues.

**Figure 8 ijms-24-14708-f008:**
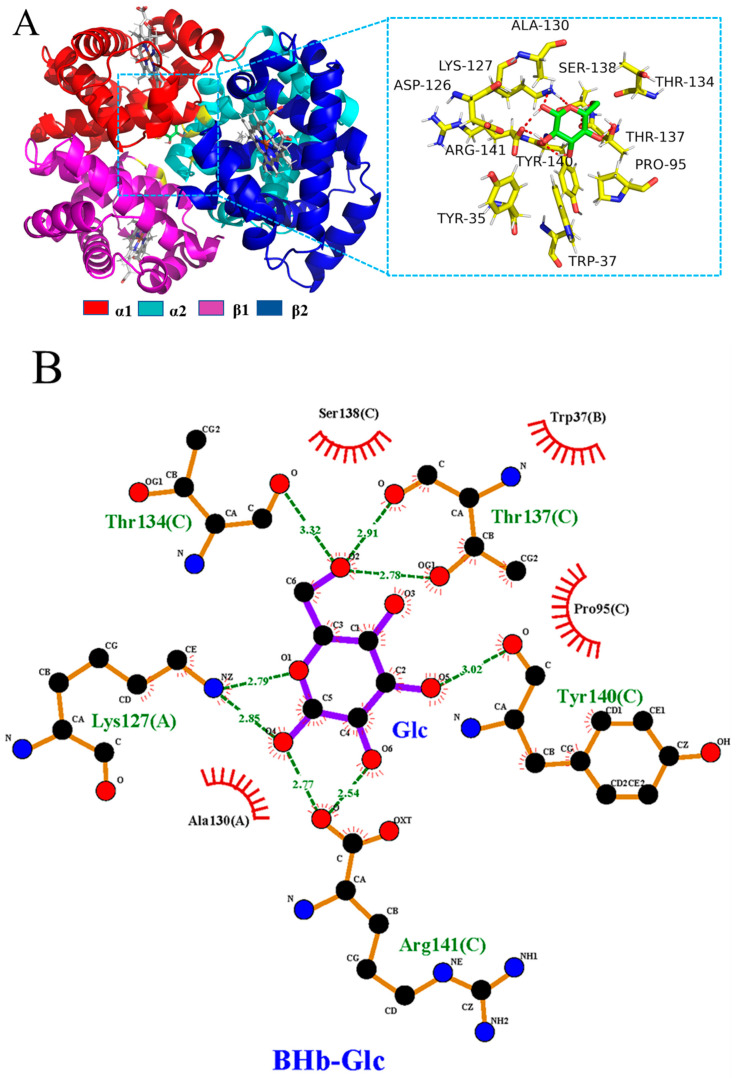
(**A**) Cartoon of the combination of BHb and Glc. And the amino acid residues of BHb surrounding Glc within 4.00 Å (red dashed lines represent hydrogen bonds). (**B**) The 2D molecular interaction plot for Glc surrounded by contacting receptor residues.

**Figure 9 ijms-24-14708-f009:**
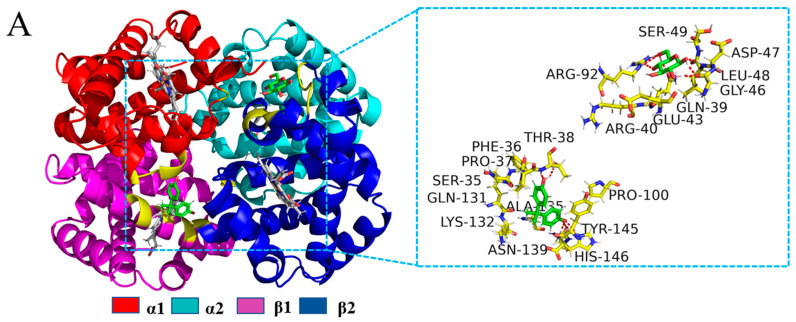
(**A**) Cartoon of the combination of BHb and BPA–Glc. And the amino acid residues of BHb surrounding BPA–Glc within 4.00 Å (red dashed lines represent hydrogen bonds). (**B**) The 2D molecular interaction plot for BPA–Glc surrounded by contacting receptor residues.

**Table 1 ijms-24-14708-t001:** Percentage of α-helices calculated for BHb, BHb + Glc, BHb + BPA and BHb + BPA + Glc.

	BHb	BHb + BPA	BHb + BPA + Glc	BHb + Glc
α-Helix%	28.7	32.4	33.9	29.7

**Table 2 ijms-24-14708-t002:** Percentage of secondary structures of BHb, BHb + Glc, BHb + BPA and BHb + BPA + Glc calculated by FTIR detection.

	α-Helix% (1660–1649)	β-Sheet% (1637–1610)	β-Turn% (1680–1660)	Antiparallel β-Sheet% (1690–1680)	Random Coil% (1648–1638)
BHb	22.0	32.8	16.0	7.8	21.4
BHb + Glc	23.4	29.8	16.2	7.6	22.9
BHb + BPA	25.0	29.2	17.7	8.4	19.6
BHb + BPA + Glc	26.9	26.1	17.1	7.6	22.3

## Data Availability

Data available upon request.

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
