# Peer review of "The Effect of Glucose on the Interaction of Bisphenol A and Bovine Hemoglobin Characterized by Spectroscopic and Molecular Docking Techniques"

_ijms, 2023, doi:10.3390/ijms241914708_

Round 1
Reviewer 1 Report
In this manuscript, Xianheng Li et. al. report the effect of glucose on the interaction of bisphenol A and bovine hemoglobin. The authors use a combination of UV spectroscopy, FTIR, CD, and molecular docking to evaluate the impact of glucose on bisphenol A and bovine hemoglobin binding. This study addresses fundamental questions regarding the change in the structure of bovine hemoglobin after binding to bisphenol A. However, I have a few minor concerns regarding the CD experiments as well as data representation. Below are the point-wise concerns in detail.
Minor concerns:
1. In the CD experiments, authors have not recorded the CD spectra at the wavelength of 195 nm, which is important because this particular wavelength also provides details about the alpha-helix and it may change the overall interpretation.
2. Overall regarding the data representation, the font size of the text should be increased to all the figures for clear visualization.
3. In line 31, please insert reference at the end of the sentence.
4. In line 49, there should be space between twofold.
5. In line 54, degree should be superscript.
6. In line 138 “Ais that Glc ……binding to Bhb”. What is the Ais?
Reviewer 2 Report
Miao and his coworkers have used spectroscopic and molecular modelling techniques to study the interaction of bisphenol A (BPA) and glucose (Glc) to bovine hemoglobin (Bhb) and observed that Glc can interact with Bhb-BPA complex with greater effect than to Bhb ligand-free structure which may lead to the altered environment for Tyr and Trp residues in Bhb-BPA complex. Although the spectroscopic study was rigorously performed, the molecular modelling tasks have certain limitations which needs to be improved.
1. Why the authors have used different conformations of protein-ligand complexes to illustrate the binding of BPA, Glc and BPA+Glc to Bhb? In Figure 7A, the orientation of a1 and b1 is on the left and a2 and b2 on the right. This is in direct contrast to Figure 8B wherein a2 and b2 is shown on the left and a1 and b1 on the right side. Whether BPA and Glc binds to the same central hydrophobic cavity in the front and back side of the Bhb protein, respectively? This is highly confusing.
2. In Figure 9A, both BPA and Glc binds at the two different sites (or extended hydrophobic cavity) in the front side of the Bhb complex?
3. To show Glc and BPA binding at two different sites in the Bhb, the authors first docked BPA and then Glc to create BPA-Glc-Bhb complex. Why the BPA occupied different site to bind in this complex as in Figure 9AB? It is expected that BPA occupies the same position (central hydrophobic cavity) similar to the one shown in Figure 7AB. This indicates that the experiment is not performed well.
4. The Reviewer recommend applying multiple ligand simultaneous docking (PMID: 26414950) and pharmacophoric scoring (PMID: 30220049) technique using Autodock tools to check where BPA and Glc prefers to bind at the sites of Bhb?
5. The authors should present clustering of top 10 conformations and the majority of sites that each ligand occupies. This may be the most appropriate technique to understand multi-ligand binding.
Reviewer 3 Report
In the presented manuscript “The effect of glucose on the interaction of bisphenol A and bovine hemoglobin characterized by spectroscopic and molecular docking techniques” the authors have been studying the influence of BPA as a widespread toxic agent on association with BHg in the presence of glucose.
Despite the fact that the main purpose of the study is of great practical and fundamental importance, the introduction of the manuscript does not properly convey a full picture of the research conducted on this topic.
The Method section lacks of some details, as, for example, the wavelength of the excitation of Trp- and Tyr-fluorescence. However, it contains the information about Na2HPO4-12H2O reagent used for the preparation of Phosphate buffer (of course, the water content in the reagent powder is not necessary to be mentioned).
Curtain conclusions the authors made in Results and discussion section call into questions. For example, the authors suggest the reason of the increasing absorbance peak at 280 nm when the BPA was added to BHg is the “changes in the aromatic amino acid microenvironment in the protein” (line 125). Referring to the chemical structure of bisphenol A, it consists of two hydroxyphenyl groups which have an absorption peak at the region of 280 nm. This fact clearly explains the cause of the increasing absorbance peak at the titration of BHg by different BPA concentrations.
Another point in general refers to all the discussions regarding to fluorescent spectra. The conclusions made on the blue-shifted peaks of Trp-fluorescence don't look convincing. Since the spectral measurements were conducted with 3- and 5-nm width of the slits, the shifts from 339 nm to 337 or 335 nm could not be adequately estimated and further applied as proofs of the suggested hypotheses. Moreover, the raw data presented as units (a.u.) certainty should be normalized to percentages.
Due to aforementioned notes, it is recommended the authors thoroughly revise the methodology of the experiments and the manuscript as well.
Round 2
Reviewer 2 Report
The authors have addressed all my comments satisfactorily.
Author Response
Thank you very much for your comment!
Reviewer 3 Report
Revised version of the presented manuscript contains supplemented method description, corrected figures and conclusions. Authors agreed with reviewer’s comments and added certain explanations as it was recommended.
Nevertheless, some points again call into questions. Authors added the sentence within lines 152–157 concerning plausible reasons of increasing fluorescence intensity of BHb in presence of BPA. However, the one of possible causes of this fact could be easily examined by the measurements of absorbance of BPA alone, without BHb. It is recommended to present additional figure with absorbance spectra of the same concentrations of BPA as in the experiment on Fig. 1A, for example, as a Supplement file.
The reviewer notes regarding the water content of the reagent powder implied the excessive information on reagent description. So, the correct sentence could look like as: “The pH was controlled using 0.1 M phosphate buffer (a mixture of NaCl, KCl, KH2PO4, and Na2HPO4), KH2PO4 and Na2HPO4 were of analytically pure grade and were purchased from Sinopharm Co. Ltd.”
